# The structural basis for CD36 binding by the malaria parasite

Fu-Lien Hsieh[1], Louise Turner[2], Jani Reddy Bolla[3], Carol V. Robinson[3], Thomas Lavstsen[2] & Matthew K. Higgins[1]

CD36 is a scavenger receptor involved in fatty acid metabolism, innate immunity and angiogenesis. It interacts with lipoprotein particles and facilitates uptake of long chain fatty acids. It is also the most common target of the PfEMP1 proteins of the malaria parasite, *Plasmodium falciparum*, tethering parasite-infected erythrocytes to endothelial receptors. This prevents their destruction by splenic clearance and allows increased parasitaemia. Here we describe the structure of CD36 in complex with long chain fatty acids and a CD36-binding PfEMP1 protein domain. A conserved hydrophobic pocket allows the hugely diverse PfEMP1 protein family to bind to a conserved phenylalanine residue at the membrane distal tip of CD36. This phenylalanine is also required for CD36 to interact with lipoprotein particles. By targeting a site on CD36 that is required for its physiological function, PfEMP1 proteins maintain the ability to tether to the endothelium and avoid splenic clearance.

[1] Department of Biochemistry, University of Oxford, South Parks Road, Oxford OX1 3QU, UK. [2] Centre for Medical Parasitology, Department of International Health, Immunology & Microbiology, University of Copenhagen and Department of Infectious Diseases, Rigshospitalet, Copenhagen 1017, Denmark. [3] Physical and Theoretical Chemistry Laboratory, University of Oxford, South Parks, Oxford OX1 3QZ, UK. Correspondence and requests for materials should be addressed to M.K.H. (email: matthew.higgins@bioch.ox.ac.uk).

The majority of human deaths caused by malaria result from infection by *Plasmodium falciparum*[1]. Unlike other human-infective *Plasmodium* species, this parasite contains a set of adhesive proteins, the PfEMP1, which are expressed on the surfaces of infected red blood cells, causing them to bind to blood vessels and tissues[2]. This protects the parasite within from clearance by the spleen, allowing the development of increased parasite load[3,4]. It also leads to specific pathology, with PfEMP1 causing the accumulation of parasite-infected erythrocytes in the placenta during pregnancy-associated malaria[5,6] and in the brain during cerebral malaria[7,8].

The PfEMP1 are among the few parasite proteins constantly exposed to the host immune system during the blood phase of the parasite life cycle. They are therefore under conflicting selection pressure, experiencing pressure to diversify to aid immune evasion while retaining the capacity to interact with invariant human receptors to fulfil their cytoadhesive role. As a result they have expanded into a complex protein family, with around 60 members in each parasite genome and a system of antigenic variation that allows parasites to switch which PfEMP1 they expose to the immune system[2,9]. Antibodies that bind PfEMP1 are found in people from malaria endemic regions and correlate with immunity from severe disease[6,10–12].

To mediate endothelial binding, PfEMP1 proteins contain large extracellular ectodomains consisting of 2–10 copies of two domain types, the Duffy-binding like (DBL) and cysteine-rich interdomain region (CIDR) domains[9,13]. While these domains are extremely variable in sequence, they share a conserved architecture[14] and can be classified into different subgroups based on sequence[15]. The vast majority of PfEMP1 proteins contain a DBL domain at the membrane-distal end, followed by a CIDR domain[2], with a variable number of succeeding domains. PfEMP1 are frequently modular, with individual domains maintaining the capacity to bind to specific receptors[14]. The domains closest to the membrane-distal end interact with endothelial receptors[16–18], while serum components such as IgM[19,20] and α2-macroglobulin[21] interact with the membrane-proximal domains.

While a variety of ligands have been implicated in binding to PfEMP1, there are three major human endothelial proteins that have been shown to bind in a consistent manner that is predictable based on the sequences of their constituent domains[2]. The scavenger receptor, CD36 binds to many classes of CIDRα domain (CIDRα2-6)[22,23] and is the host receptor most commonly found to interact with parasite isolates from patients[24–26] and with laboratory-adapted parasite strains[27]. ICAM-1 is also a common partner, with binding capacity retained in a subset of the DBLβ domains[16,28,29]. Finally, most CIDRα1 domains bind to endothelial protein C receptor (EPCR) with 5-7 such domains predicted in the genome of each parasite isolate[2,17].

Approximately 84% of PfEMP1 proteins contain domains predicted to bind to CD36, making this the most common adhesion phenotype[2]. CD36 is a cell surface scavenger receptor that plays a role in fatty acid uptake, angiogenesis and phagocytosis[30]. It contains two transmembrane helices, between which lies a ~47 kDa extracellular domain. The structure of LIMP-2, a homologue of CD36, revealed an oval-shaped ectodomain with a hydrophobic cavity that traverses the whole length of the molecule. This cavity is thought to be the site through which fatty acids pass as they move towards the plasma membrane, although no fatty acids are present in the existing structures[31]. An α-helical 'platform' at the membrane distal tip of CD36 has been identified as a recognition site for oxidized low density lipoprotein (LDL) and fatty acids, with a lysine residue (K164) on this platform cross-linking to fatty acids and important for oxidized LDL binding[31,32]. The structure of CD36 and its binding surface for PfEMP1 were unknown, although a peptide-based study identified residues 146–164 of CD36 as forming the binding site[33].

CD36 is present in many tissues, allowing adhesion of infected erythrocytes in different sites within the vascular endothelium[18]. It is also present in the cells of the innate and acquired immune systems, modulating the binding of infected erythrocytes to macrophages and dendritic cells[34,35]. This leads to a nuanced interaction between parasite and host[36]. On one hand, the frequency of CD36 binding shows that it is advantageous to the parasite, allowing the avoidance of splenic clearance. CD36 binding may also benefit the parasite by providing a mechanism to reduce dendritic cell-mediated T-cell activation, hampering the capacity of the host immune system to clear the infection[34,37]. On the other hand, CD36 binding increases the likelihood of infected erythrocytes being cleared by macrophages, perhaps reducing the chances of developing high parasitaemia[35,38]. CD36 binding is therefore thought to allow the parasite to sustain a viable infection, while not putting the host at significant danger from elevated parasite levels[36].

The role of CD36 binding in malaria has been studied extensively. However, significant sequence diversity in CD36-binding CIDR domains has hampered efforts to understand the molecular determinants that are retained to allow interaction with CD36. To understand what makes CD36-binding such a successful interaction for the parasite, we have therefore combined structural and biophysical studies with analysis of thousands of sequences to show how CIDRα domains interact with CD36. We find that, despite extensive sequence diversity, the CD36-binding CIDRα domains retain a conserved hydrophobic pocket that accepts a phenylalanine residue found at the membrane distal tip of CD36. This phenylalanine is also required for CD36 to bind to oxidized LDL particles. We therefore show that the diverse PfEMP1 proteins retain a chemically conserved surface feature that allows them to interact with an invariant region of their receptor, maintaining the ability to tether to the endothelium and to avoid clearance.

## Results

**Determining the structure of CD36 bound to a CIDRα domain.** To determine the structure of CD36 in complex with a CIDRα domain, we selected the CIDRα2.8 domain from the MCvar1 PfEMP1, as the structure of this isolated domain was already known[39]. We coexpressed the ectodomain of CD36 (residues 35–439) together with the MCvar1 CIDRα2.8 domain (residues 576–745 with a C626S mutation to remove an unpaired cysteine). These were expressed in human embryonic kidney (HEK) 293 cells in the presence of the class I α-mannosidase inhibitor, kifunensine and the secreted complex was purified. We used size-exclusion chromatography coupled to multi-angle light scattering (SEC-MALS) to demonstrate the formation of a 1:1 complex, with no higher-order protein assemblies (Supplementary Fig. 1).

The CD36:CIDRα2.8 complex was subjected to crystallization trials with *in situ* deglycosylation and proteolysis, allowing a complete data set to be collected to 2.07 Å resolution. Molecular replacement attempts in which the MCvar1 CIDRα2.8 domain (PDB 3C64)[39] was used as a search model failed, while using LIMP-2 (PDB 4F7B)[31] as a search model yielded a solution (Supplementary Fig. 2). Subsequent analysis of the electron density revealed that the MCvar1 CIDRα2.8 domain structure had adopted a very different conformation from that seen in crystals of the domain alone (Supplementary Fig. 3). The molecular replacement solution obtained using LIMP-2 was therefore used as the starting point for an iterative cycle of model building and refinement to generate a structure of the CD36:CIDRα2.8 complex (Table 1).

**The structure of CD36 bound to fatty acids**. The CD36 ectodomain adopts an architecture similar to LIMP-2, with an elongated structure, and is decorated with nine N-linked glycans distributed across the protein surface (Fig. 1a, Supplementary Fig. 4)[31]. While LIMP-2 undergoes a pH-dependent conformational change, with an 'open' conformation at pH 5.5 and a 'closed' conformation at pH 7.5 (refs 31,40,41), CD36, which was crystallized at pH 8.0, adopts an open conformation (Fig. 1b). Indeed the pH-sensor residue, H150 of LIMP-2, which stabilizes its closed conformation[41] has been replaced by phenylalanine residue (F153) in CD36, and CD36 will therefore not undergo the pH-driven conformational change observed in LIMP-2, but will remain open and poised to accept fatty acids at neutral pH.

A distinctive feature of LIMP-2 is a cavity that spans most of the length of the molecule[31]. This is also present in CD36 and is lined predominantly by hydrophobic side chains. Unlike in LIMP-2, the cavity of CD36 contains two electron density features that resemble the extended hydrocarbon chain of a fatty acid. We identified these fatty acids by chloroform extraction of ligands from the crystallized protein, followed by GC/MS of the extracted material. This revealed that palmitic acid and stearic acid were most prevalent of a mixture of fatty acids present in the structure (Supplementary Fig. 5). Indeed CD36 has been shown to play a role in the uptake of these fatty acids[42] and they have most likely bound to the protein during expression in human culture cells. We therefore modelled palmitic acid into these electron densities and observed a close fit. Both fatty acids are oriented with their carboxylic acid moieties at the membrane distal end. The majority of their interactions with CD36 are mediated by hydrophobic contacts as the fatty acids lie along the length of the cavity. However, the carboxylic acid group of the

membrane distal fatty acid also makes water-mediated hydrogen-bonding interactions with T195.

The entrance for the fatty acid translocation pathway has previously been proposed to lie close to K164 (refs 31,32). Indeed the central cavity has an opening close to this residue (entrance 1) that is a likely entry point for the membrane distal fatty acid and is the entrance that opens and closes in response to pH changes in LIMP-2 (Fig. 1c). In addition, a second opening (entrance 2) is also found at the membrane distal side of the CD36 ectodomain (Fig. 1c) and the second fatty acid occupying the cavity lies in a tunnel that proceeds from this entrance. Our CD36 structure therefore supports a model in which fatty acids move through the cavity, identifying a second possible entrance and further delineating favourable positions through which translocating

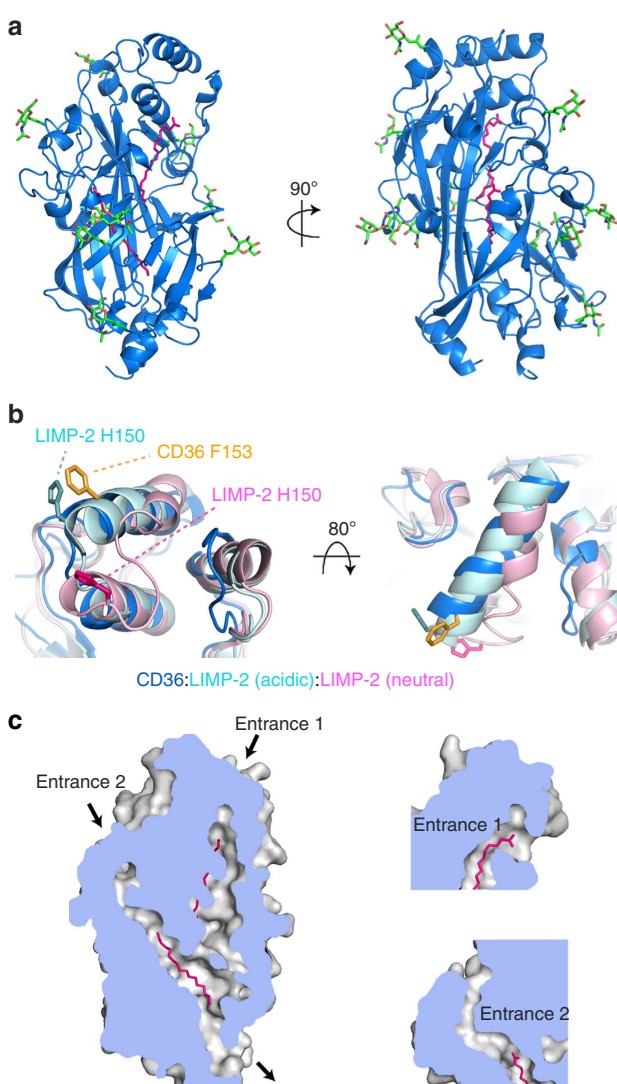

**Figure 1 | The structure of CD36 and its binding of fatty acids.** (**a**) The structure of CD36, shown in blue. The nine N-linked glycosylation sites and associated sugars are green while two palmitic acids are shown as pink sticks. (**b**) An alignment of CD36 (blue) with the structures of LIMP-2 at acidic (cyan) and neutral (pink) pH. Residues F153 from CD36 (orange) and H150 from LIMP-2 at acidic (light blue) and neutral (pink) pH are highlighted. (**c**) A section through a surface representation of CD36 showing the central core cavity occupied by two palmitic acids (pink). Insets show two putative entrances to this central cavity.

**Table 1 | Data collection and refinement statistics.**

| | CD36:CIDRα2.8 |
|---|---|
| *Data collection* | |
| Space group | C121 |
| *Cell dimensions* | |
| a, b, c (Å) | 129.8, 40.7, 138.8 |
| α, β, γ (°) | 90.0, 114.9, 90.0 |
| Wavelength | 0.97949 |
| Resolution (Å) | 63.00–2.07 (2.12–2.07) |
| $R_{pim}$ (%) | 4.6 (53.1) |
| I/σI | 8.2 (1.5) |
| Completeness (%) | 99.8 (99.5) |
| Multiplicity | 3.2 (3.3) |
| | |
| *Refinement* | |
| Number of reflections | 40,525 (3,984) |
| $R_{work}/R_{free}$ | 21.2/25.2 |
| *Number of atoms* | |
| Protein | 4,490 |
| Ligands | 233 |
| Water | 152 |
| *B-factors* | |
| Protein | 47.6 |
| Ligand/ion | 64.4 |
| Water | 44.6 |
| *R.m.s deviations* | |
| Bond lengths (Å) | 0.01 |
| Bond angles (°) | 1.19 |
| *Ramachandran plot* | |
| Favoured (%) | 97.0 |
| Allowed (%) | 3.0 |

All structures were determined from one crystal.
Values in parentheses are for highest-resolution shell.

ligands move as they pass towards the membrane proximal exit and the plasma membrane.

**The structural basis for CD36 binding by PfEMP1.** As expected from characterizing the stoichiometry of the CD36:CIDRα2.8 complex (Supplementary Fig. 1), the crystals contained an equal number of CD36 and CIDRα2.8 molecules. However, two CIDRα2.8 domains make significant contacts with each CD36 ectodomain. To determine which of these represents the physiological interaction, and which is due to crystal packing we assessed the binding of mutant proteins by surface plasmon resonance. Glycosylation sites were introduced into the CIDRα2.8 domain to disrupt each of the two interfaces. In addition, a key interaction at site 1 is mediated by the insertion of a phenylalanine residue (F153) from CD36 into a hydrophobic pocket on the CIDRα2.8 domain and we therefore produced the F153A mutant of CD36. The insertion of a glycosylation site into site 1 and the F153A mutation both disrupted the binding of the CIDRα2.8 domain to CD36 (Supplementary Fig. 6). In contrast, a mutation inserting a glycan into site 2 had no effect on the interaction, identifying site 1 as the authentic binding surface.

The PfEMP1 CIDRα2.8 domain therefore binds to the α-helical bundle at the membrane-distal tip of CD36, close to the major entrance (entrance 1) to the hydrophobic cavity (Fig. 2a) and in the region identified previously[33]. The majority of the interaction is mediated by F153 from CD36, which fits into a hydrophobic pocket on the CIDRα2.8 domain surface. Mutation of either F153, or of hydrophobic residues of the CIDRα2.8 domain that line this pocket, abolishes the interaction (Fig. 2b, Supplementary Fig. 7). Secondary interactions are made by a smaller α-helix (residues 668–678) from the CIDRα2.8 domain (mediating hydrogen bonding to CD36 through E672) and a loop (residues 647–656; again hydrogen bonding, this time through D650). Neither of these interactions is critical in the MCvar1 CIDRα2.8 domain, as disrupting hydrogen bonds by mutagenesis led only to small reductions in affinity (Fig. 2b, Supplementary Fig. 7, Supplementary Table 1).

Previous studies have shown reduced binding of infected erythrocytes to CD36 in response to phosphorylation of T92 (ref. 43). We see no evidence of phosphorylation of T92 in the electron density maps and this side chain is >40 Å from the CIDRα2.8 domain, suggesting that T92 phosphorylation will not directly block the interaction.

Four structures of CIDRα domains have been solved to date[39,44,45]. Of these, three adopt a similar compact conformation[44,45], while the MCvar1 CIDRα2.8 domain adopts an open conformation when crystallized at pH 4.2 (ref. 39) (Supplementary Figs. 3 and 8). In the structure presented here, the MCvar1 CDRα2.8 domain (crystallized at pH 8.0) adopts the typical compact configuration found in other CIDR domains crystallized around physiological pH, and aligns to the CIDRα1 and γ domains with rmsd of 3.4 and 4.3 Å respectively (Supplementary Fig. 3). To understand the reasons for this structural discrepancy, we crystallized our HEK293 cell expressed MCvar1 CIDRα2.8 domain and obtained crystals at pH 4.5 with the same space group and close to identical cell dimensions (p6₅22; $a = b = 95.9$ Å, $c = 91.9$ Å) to those reported for bacterially expressed protein crystallized at pH 4.2. We therefore propose that the open conformation is an artefact of crystallizing the isolated CIDRα2.8 domain in acidic conditions and that the domains adopt a conserved compact architecture under physiological conditions.

A comparison of the molecular features used by CIDRα domains to bind to CD36 and EPCR shows striking similarities, but with a structural inversion of the mechanism of binding (Fig. 3). In both cases, the CIDRα domains contain a core three α-helical bundle with an insertion between the second and third core helices, α2 and α6 (Supplementary Fig. 8). This folds to make smaller helices that lie approximately perpendicular to the core helices. This insert has been identified as a 'homology block' and CIDR domains can be classified into three groups, the CIDRα1 domains, the CIDRα2-6 domains and the CIDRβ/γ/δ domains, which each have a different homology block in this region[15]. Now that we have structures for members of each of these domain classes (Supplementary Fig. 3) we can see how these homology blocks fold to contribute to the formation of ligand-binding sites.

In the CIDRα1 domains, the insert folds to form a helix and a kinked helix, with a critical phenylalanine residue protruding from the kink. This hydrophobic protrusion forms the centre of the binding site, occupying the hydrophobic groove of EPCR. It is surrounded by hydrogen bonding residues that further stabilize the interaction. In contrast, CIDRα2-6 domains lack the kinked helix and the protruding phenylalanine. Instead the equivalent region of the protein contributes, together with residues from the core three-helical bundle, to a hydrophobic cavity (Fig. 3), which accepts a protruding phenylalanine residue from the ligand (F153 from CD36). Here too, the hydrophobic core of the interaction surface is surrounded by hydrophilic regions that can make hydrogen bonds. Residues previously identified to distinguish between CD36-binding and non-binding CIDR domains lie within this homology block, but do not directly contact CD36 (ref. 46). Therefore in both cases the domains form an extremely stable interaction using a contact surface built around a small hydrophobic core, surrounded by hydrogen bond donors and acceptors. The insertion of different homology blocks into the core helical bundle of the CIDRα domains allows them to do this in different ways, presenting ligand-binding surfaces with different architectures to interact with differently structured ligands.

**Conserved structural features allow CD36 binding.** We next aimed to understand how the CD36-binding CIDRα2-6 binding domains diversify, while retaining ligand binding. The genome sequence of the 3D7 strain of *Plasmodium falciparum* allowed a previous systematic analysis of PfEMP1 to identify those that contain CIDR domains that interact with CD36 (ref. 23). Three classes of CIDR domains were identified, CIDRα, β and γ. All tested CIDRα domains, with the exception of those from the CIDRα1 subgroup, bound to CD36 (ref. 23) while CIDRα1 domains bind to EPCR (ref. 17). This suggested that CD36 binding is an extremely common phenotype, with ∼85% of PfEMP1 proteins containing the CIDRα2-6 domains predicted to bind. However the CIDRα2-6 domains are highly variable with less than 33% pairwise sequence similarity[2].

With many PfEMP1 sequences emerging from recent parasite genome sequencing projects, we were able to conduct a more comprehensive analysis. We built sequence distance trees using 2386 CIDRα2-6 domain sequences extracted from 233 parasite genomes that had been selected to be representative of global diversity[15]. No major sequence clusters could be verified by bootstrapping, but the sequences distributed into small subgroups as previously described (Fig. 4a)[15]. The alignment included 11 CIDRα domains from *Plasmodium reichenowi* that distributed across the tree of *Plasmodium falciparum* CIDRα2-6 domains, as seen for other surface protein families[47]. We used surface plasmon resonance to test a panel of CIDRα2-6 domains randomly selected from across this sequence diversity tree for the ability to bind to CD36, showing them all to adopt nanomolar interactions with slow off-rates (Fig. 4a, Supplementary Fig. 9). Together with previously identified CD36 binders[17,23,27], this confirmed that all tested members of the CIDRα2-6 classes bind to CD36.

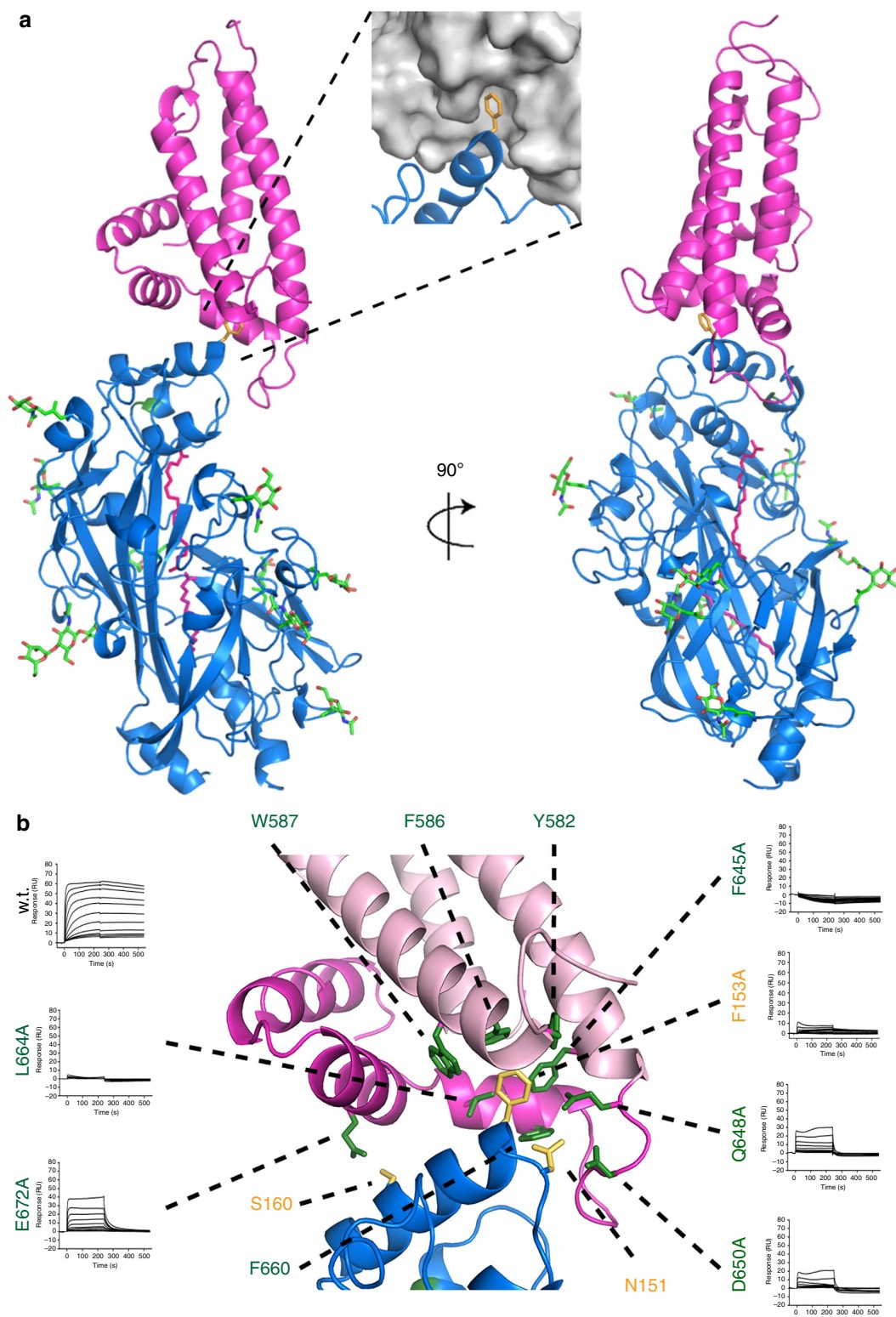

**Figure 2 | The structural basis for CD36 binding by PfEMP1 proteins.** (**a**) The structure of the complex of CD36 (blue) with the CIDRα2.8 domain of the MCvar1 PfEMP1 protein (pink). *N*-linked glycans are shown in green. The inset shows a surface representation of the CIDRα domain with CD36 as a blue cartoon and F153 of CD36 in orange. (**b**) A close-up view of the interface between CD36 and the CIDRα domain with the key interacting residues labelled. The three-core α-helices of the CIDRα domain (α1, α2 and α6) are coloured in light pink while the insert (α3-α5) is dark pink. Also shown is the effect of mutagenesis of key interacting residues as determined by surface plasmon resonance. The most critical interactions are mediated by F153 of CD36, which fits into a hydrophobic pocket lined by residues including F645 and L664 of the CIDRα domain.

We next produced a sequence logo for all members of these domain subclasses (Fig. 4b,c). The majority of the domain sequences could be aligned into a single logo. However, the loop containing residues 673-683 was divergent in the CIDRα3.1 subclass. As found before in the DBL and CIDR domains, the majority of conserved residues were cysteines or aromatics[14,44].

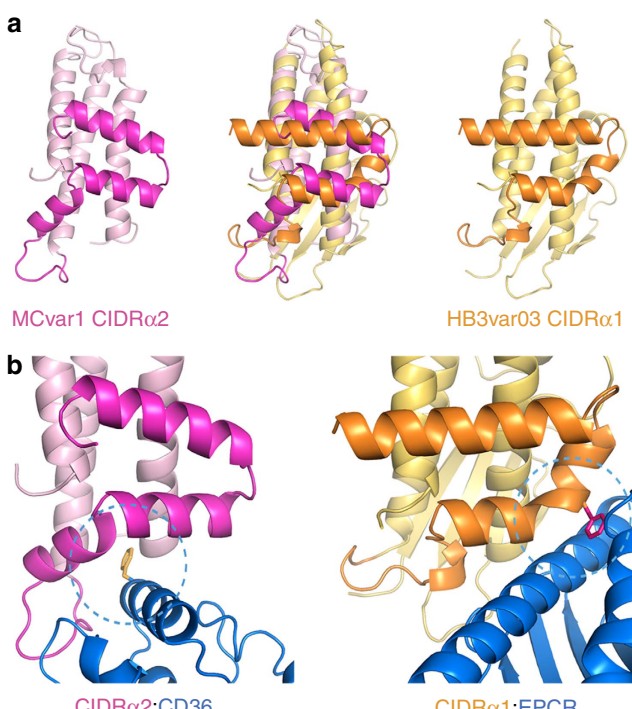

**Figure 3 | A comparison of the features that allow CIDRα domains to bind to EPCR or CD36.** (**a**) A comparison of the structure of a CD36-binding CIDRα2 domain (pink) with that of an EPCR-binding CIDRα1 domain (orange). Both domains share a core three α-helical bundle. The insertion that emerges between the second and third of these core helices forms a docking platform for ligands. (**b**) A close up of the binding interfaces that mediates the CIDRα2:CD36 and CIDRα1:EPCR interactions. The CIDRα1 domains have a phenylalanine residue (F656) on a convex surface of the domain that protrudes into the hydrophobic groove of EPCR. In contrast, the CIDRα2 domains have a hydrophobic pocket that binds to a protruding phenyalanine residue (F153) from CD36.

A sequence logo for residues that make direct interactions with CD36 (Fig. 4d) showed limited sequence conservation, but with hydrophobic residues exchanged for other hydrophobic residues and hydrogen bonding capacity conserved. Three of the most conserved residues that directly contact CD36 (Y582, F586 and F645) contribute to the hydrophobic pocket that accepts F153 (Fig. 2b).

To understand the distribution of conservation across the CD36-binding domains, we plotted sequence entropy, as a measure of chemical conservation, onto the structure of the MCvar1 CIDRα2.8 domain (Fig. 5a,b). As seen previously for the CIDRα1 domains, the significant majority of conserved residues were found in the core of the domain, stabilizing the protein fold. The major patch of surface exposed conservation was in, and immediately surrounding, the conserved hydrophobic patch, with substitutions in this region leading to conservation of the hydrophobicity of this pocket, maintaining the ability to accept the conserved F153 residue from CD36. Other contacting residues lacked direct conservation, but retained the ability to form hydrogen bonds. This suggests a shared binding mode for CD36 across the CIDRα2-6 domains.

To support this conclusion, we tested the effect of the F153A mutation of CD36 on the binding of a sequence diverse set of ten CIDRα2-6 domains. We found that the binding of each of these domains to CD36 was significantly reduced in the presence of the F153A mutations, with reduced overall binding levels and significantly faster off rates (Fig. 5c, Supplementary Fig. 10).

The F153A mutation did not have an equivalent effect on all of the CIDRα domains, suggesting that the importance of additional interactions differ across the family. However, conservation of chemical property in the F153 binding pocket, combined with a significant effect of F153 mutation on all of the interactions supports the idea of a conserved binding mode across the domain family, despite significant sequence diversification.

**The effect of CIDR domains on the natural function of CD36.** CD36 has a number of functions in mammalian physiology. It has the ability to bind oxidized lipoprotein particles[31] and to interact with thrombospondin (TSP) to regulate angiogenesis[30,48]. We therefore investigated whether the binding of PfEMP1 to CD36 can prevent it from interacting with physiological binding partners. Surface plasmon resonance experiments showed that the binding of CD36 to TSP was not altered by pre-incubation with CIDRα2-6 domains (Fig. 6a). In contrast, the presence of CIDRα2-6 domains abolished the interaction of CD36 with oxidized LDL particles (Fig. 6b,c). Furthermore, the F153 mutation, which disrupts the interaction of CD36 with CIDRα2-6 also abolishes binding of CD36 to oxidized LDL particles, showing that this region of the protein is essential to interact both with lipoprotein particles and PfEMP1 (Fig. 6d). The parasite therefore targets a part of CD36 that is essential for its physiological role in fatty acid uptake, reducing the likelihood that the human host can escape from PfEMP1 binding by altering its CD36.

## Discussion

The PfEMP1 proteins of *Plasmodium falciparum* have evolved under conflicting evolutionary pressures. On one hand, they are constantly exposed to the immune system and are therefore under constant pressure to diversify, allowing immune evasion. On the other hand, they must retain functionally important features, maintaining the capacity to interact with specific endothelial ligands, allowing them to perform their role in endothelial adhesion. In this study we have combined structural studies of a MCvar1 CIDRα2.8 domain in complex with CD36 with the analysis of the sequences that represent much of the global diversity of CD36 binding PfEMP1 domains, in order to identify the molecular features that are required to mediate CD36 binding.

The CIDR domains provide a versatile molecular scaffold for the evolution of different binding phenotypes. Their structure is built upon a core three-helical bundle and between the second and third core helices emerges an insertion of ∼60 residues. Much of this insertion has been identified as a 'homology-block', and is a module that differs between the main CIDR domain classes[15]. In the CIDRα1 domains the insertion is ∼60 residues in length and folds to form the helix and kinked-helix that provide the majority of the EPCR-binding surface[44]. In the CIDRα2-6 domains, a different insertion in this region, of ∼70 residues, adopts a subtly different structure. The helix and kinked-helix of CIDRα1 domains are replaced in CIDRα2-6 domains by two shorter helices (α4 and α5 in Supplementary Fig. 8). Together with residues emerging from the core three-helical bundle, these form an open, hydrophobic cavity containing conserved aromatic residues, which is missing in the CIDRα1 domains. It is this cavity that forms the majority of the interaction site for CD36. Therefore, the equivalent regions of CIDRα domains adopt very different architectures, providing binding sites for very different ligands.

Although the binding sites for EPCR and CD36 are very different in shape, both interactions are mediated by similar chemical properties. In the case of EPCR binding, the CIDRα1 domains present a convex surface formed from two hydrophobic

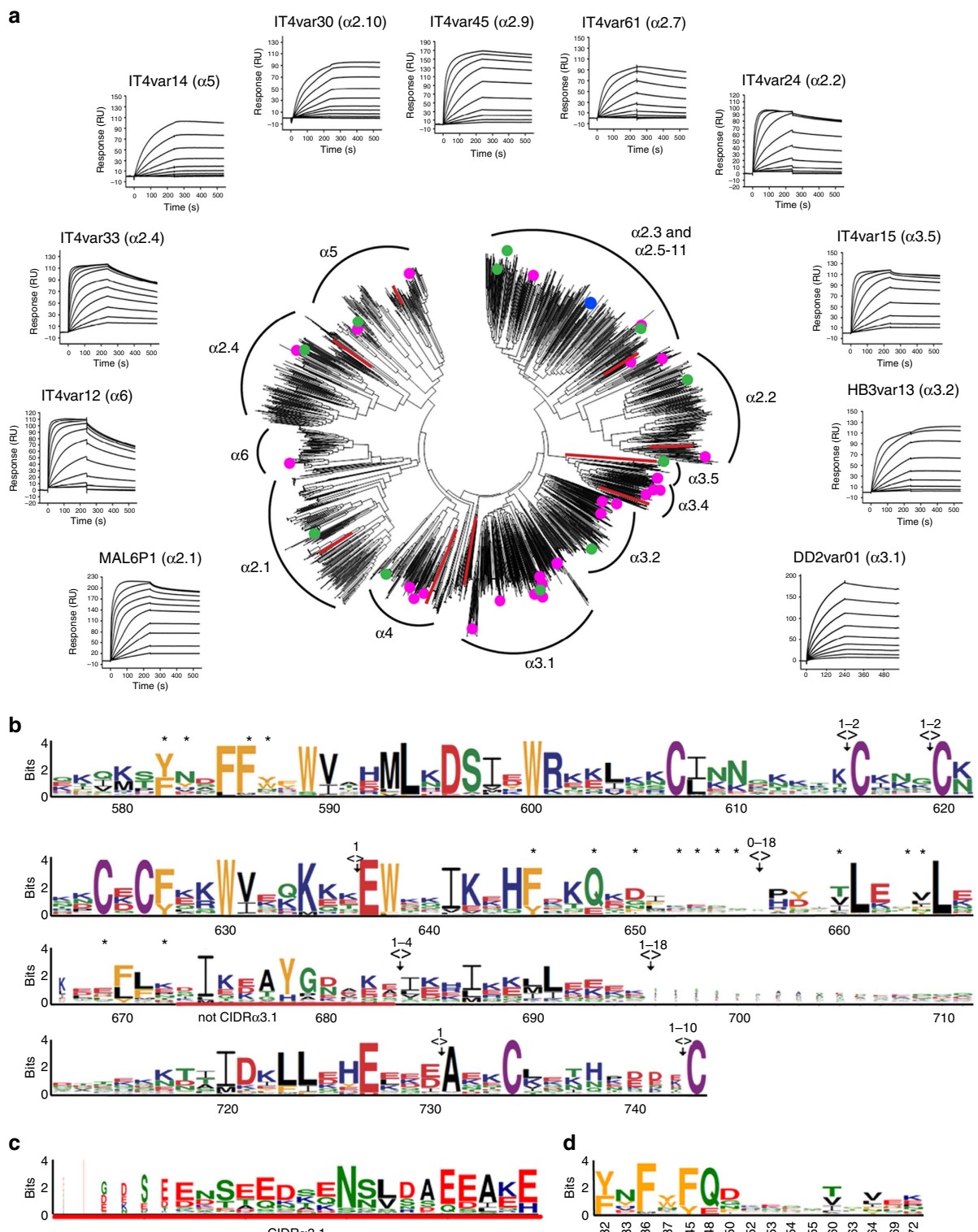

**Figure 4 | Extensive diversity in CD36 binding CIDRα domains.** (**a**) Sequence distance tree of 2386 full-length CIDRα2-6 domains. Red lines represent sequences from *Plasmodium reichenowi*. A blue circle marks the sequence of the crystallized CIDRα2.8 domain (also see Supplementary Fig. 8). Green circles mark sequences of recombinant CIDRα domains for which the affinity for CD36 binding were tested (corresponding SPR traces given). Pink circles mark sequences of CIDR domains previously demonstrated to bind CD36. Annotated clusters contained previously defined CIDRα2-6 subclasses. All tested domains from the CIDRα2-6 subclasses bind to CD36. (**b**) Sequences of the 2386 CIDRα domains were aligned and a sequence logo generated of residues equivalent to those found in the MCvar1 CIDRα domain (numbered as in MCvar1). Deletions ( > < ) and insertions ( < > ) are indicated. The region underlined by a red line is found in all CIDRα2-6 domains except for CIDRα3.1. Residues labelled with * make direct contact to CD36. (**c**) The region found in CIDRα3.1 domains that replaces that underlined in red. (**d**) A sequence logo for the residues that make direct contacts to CD36.

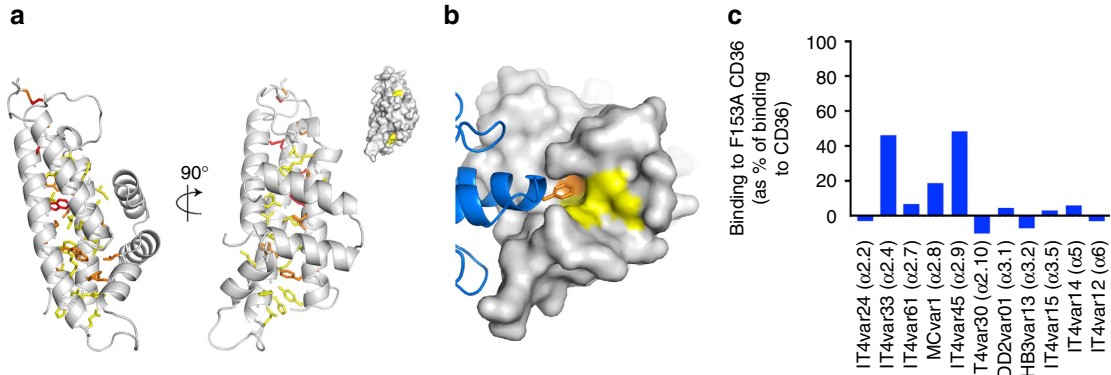

**Figure 5 | Limited chemical conservation allows CD36 binding.** (**a**) Conservation in the CD36-binding CIDRα domain is plotted onto the structure of the MCvar1 CIDRα2.8 domain. Absolutely conserved residues are shown as red sticks. Residues with property entropy score of less than 0.1 (but not totally conserved) are orange and those with scores of 0.1–0.3 are yellow. The inset shows a surface representation in the same orientation and colours, showing that conserved residues cluster predominantly in core of the domain, stabilizing its structure. (**b**) A surface representation of the CIDRα domain coloured as in A, with CD36 in blue. This shows that residues in the hydrophobic pocket of the CIDRα domain are the most chemically conserved feature on the CIDRα domain surface. (**c**) The effect of the F153A mutant of CD36 on the binding of a diverse panel of CIDRα2-6 domains shows that the interaction mediated by F153 of CD36 plays an important role in binding across the CIDRα2-6 domain family.

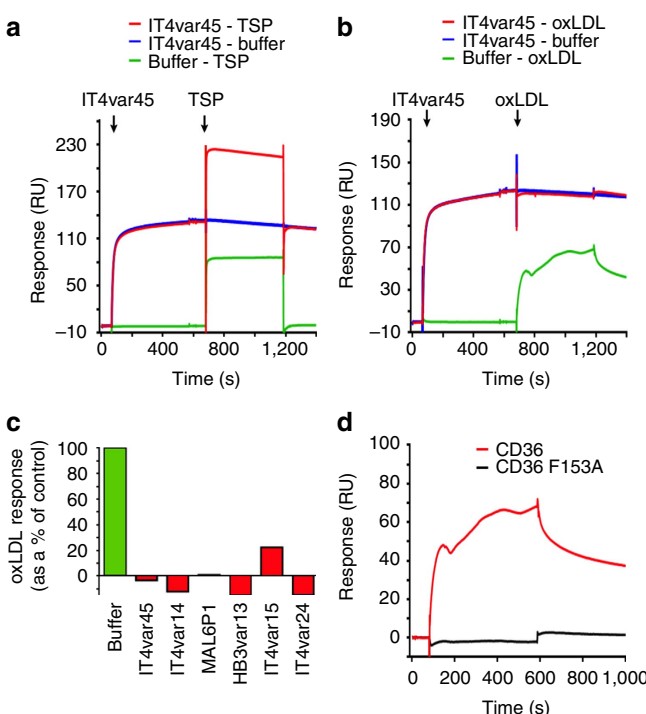

**Figure 6 | PfEMP1 proteins prevent oxidized LDL from binding to CD36.** (**a**) Surface plasmon resonance analysis showing that the preincubation of CD36 with IT4var45 CIDRα2.9 does not affect the binding of thrombospondin (TSP) to CD36. (**b**) Surface plasmon resonance analysis showing that the preincubation of CD36 with IT4var45 CIDRα2.9 prevents the binding of oxidized LDL particles (oxLDL) to CD36. (**c**) SPR data showing the inhibition of oxLDL binding to CD36 by a panel of CIDRα2-6 domains. (**d**) Demonstration by surface plasmon resonance that the F153A mutation blocks the binding of oxLDL to CD36.

phenylalanine residues. These residues, and in particular F656 in the HB3var03 CIDRα1 domain, protrude into the hydrophobic groove of EPCR, while hydrophilic residues that surround this hydrophobic cluster make hydrogen-bonding interactions. The CD36-binding domains present a binding site that is conceptually

the inverse of this. Instead of a hydrophobic protrusion, the CD36-binding surface is predominantly formed from a concave hydrophobic pocket into which docks a phenylalanine, F153 from CD36. A loop from the CIDRα domain then makes hydrogen bonds with other regions of CD36. In both cases, this combination of a small hydrophobic core, together with a larger, complementary hydrophilic interface, generates a high affinity and stable binding site, allowing the domains to interact with their ligand with a slow off rate, stabilizing cytoadhesion of infected erythrocytes against the strong forces of blood flow.

In both cases, these binding surfaces have diversified significantly, with little sequence conservation in the residues that made direct contacts with EPCR or CD36. However, in both cases, residues involved in stabilizing the fold are conserved and sequence variation in residues that interact directly with the ligands is conservative in chemistry, maintaining the capacity to bind. This raises the possibility of generating antibodies that recognize these conserved chemical features and show broadly inhibitory potential. In particular, the EPCR binding surface of the CIDRα1 domains protrudes from the domain, presenting a surface that appears compatible with recognition by an antibody. In contrast, the CD36 binding site is concave, with its conserved hydrophobic residues concealed in a pocket that may be less readily recognized. This binding surface is surrounded by a sequence diverse protein surface containing a flexible loop (residues 647–657) that might make antibody-mediated recognition less likely. Perhaps this is why the CD36 interaction is such a successful one from the perspective of the parasite, with the binding site reducing the exposure of chemically conserved determinants from detection by acquired immunity, allowing a diverse set of CIDRα2-6 domains and a system of antigenic variation to provide the changes necessary to maintain the capacity to cytoadhere and to protect the parasite from splenic clearance.

A second strategy used by both EPCR- and CD36-binding CIDR domains is to interact with functionally important sites in their human receptors to avoid the ready evolution of escape mutants in the human genome. The CIDRα1 domains bind to the region of EPCR that is required for its interaction with its ligand, protein C. Indeed the conserved phenylalanine of the CIDRα1 domains occupies a pocket that is also occupied by a phenylalanine from protein C. The parasite has evolved a similar trick in its CIDRα2-6 domains (Fig. 7). Here the CIDR domains

interact with a phenylalanine residue that protrudes from the surface of CD36. But why are polymorphisms in this residue not favoured in malaria endemic regions, as these would be expected to allow escape from binding? We show here that this residue is also important for binding to oxidized LDL particles, one of the physiological binding partners of CD36, and that the CIDRα domains prevent oxidized LDL from binding. Once again, the PfEMP1 proteins target a functionally important site, reducing the likelihood of a host escape mutant that would prevent infected erythrocytes from sequestering away from splenic destruction.

In addition, the region of CD36 on which this phenylalanine is displayed is the site of recognition for long chain fatty acids. Fatty acids can be cross-linked to K164 of CD36 and are then thought to move across to enter the central CD36 cavity and pass through the CD36 ectodomain to the membrane surface. The CIDRα2.8 domain lies directly in between K164 and the cavity entrance, potentially blocking lipid uptake. With fatty acid uptake and oxidized LDL binding linked to fatty acid metabolism and atherosclerosis, it will be interesting to see whether CIDRα2-6 domains can prevent fatty acid uptake as a proof of principle of targeting this region of CD36.

In summary, our characterization of the structural basis for CD36-binding by PfEMP1, coupled with our previous analysis of EPCR-binding has shown how the complex PfEMP1 protein family balances the need to adhere tightly to unchanging human receptors while diversifying to evade antibody-mediated recognition. Functionally important regions of human receptors are targeted, reducing the likelihood of mutations that disrupt binding, but also, inadvertently blocking the physiological roles of these receptors, with the potential to lead to disease outcomes. While human receptors are restricted from evolving away from the interaction, the PfEMP1 proteins are hugely diverse, retaining their structure and the chemical features of their binding site while altering nearly all of their surface residues. The CD36-binding CIDR domains also tuck their ligand-binding site in a small pocket, reducing the likelihood of antibody-mediated recognition. In this way, the parasite has evolved a surface protein family that can maintain the capacity to tether it away from splenic clearance, while avoiding clearance by the acquired immune system.

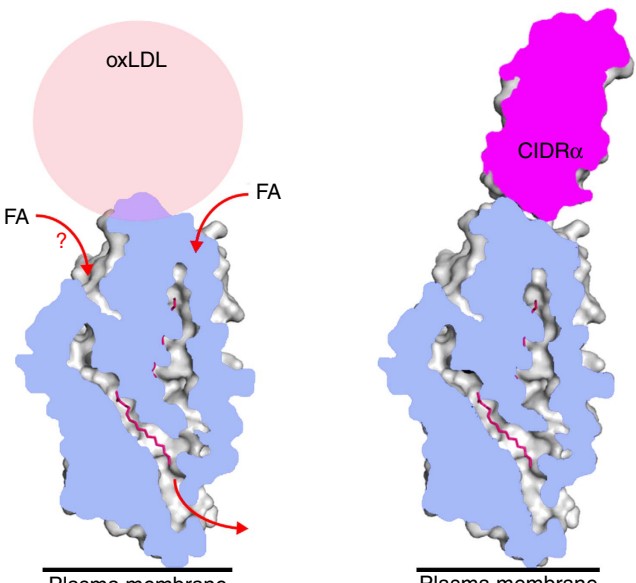

**Figure 7 | A model for the inhibition of oxLDL binding by PfEMP1.** One of the physiological roles of CD36 is to interact with oxidized LDL (oxLDL). Fatty acids can be incorporated from oxLDL particles, or other transport systems and pass through the central cavity of CD36 to the membrane. The PfEMP1 CIDRα2-6 domains interact with the same surface of CD36 as oxLDL and compete for binding.

## Methods

**Protein expression and purification.** A gene encoding the human CD36 ectodomain (residues 35–439) was inserted into the pHLsec vector[49], modified with a C-terminal 8xHis tag (pHLsec-8H). A synthetic complementary DNA clone of MCvar1 CIDRα2.8 with Cys626 mutated into Ser to remove an unpaired cysteine (codon-optimized for expression in mammalian cells) was obtained from GeneArt (Invitrogen), and residues 576–745 were cloned into pHLsec-8H. CD36 constructs used for SPR experiments had an additional C-terminal BAP tag that was biotinylated in vitro by incubation with BirA ligase (Sigma-Aldrich). Mutants were generated using two-step overlapping PCR experiments and verified by DNA sequencing (Source Bioscience) and cloning primers are given in Supplementary Table 2.

CD36 and MCvar1 CIDRα2.8 were coexpressed in HEK293 (ATCC CRL-11268) cells in the presence of 5 µM kifunensine. The conditioned media was dialysed against phosphate-buffered saline with the addition of 0.5 M NaCl and the protein complex was purified by immobilized metal ion affinity chromatography using TALON resin (Clontech). The CD36:CIDRα2.8 complex was further purified by size-exclusion chromatography using a Superdex 200 16/600 column (GE Healthcare Life Sciences) and 10 mM Bis-Tris, pH 6.5 and 0.3 M NaCl. For SEC-MALS experiments, proteins were expressed and purified as above. For surface plasmon resonance analysis, CD36 and MCvar1 CIDRα2.8 proteins and their mutants were expressed individually in HEK293 cells, and were purified as above, before buffer exchange into 10 mM HEPES, pH 7.5, 0.15 M NaCl.

Recombinant CIDRα2-6 domains were expressed and purified as previously described[44]. In short, domain sequences with boundaries as defined in Rask et al.[15] were used to design domain-encoding DNA. These were codon optimized for expression in Drosophila Sf9 cells and were delivered already inserted into the baculovirus-insect cell expression vector pAcGP67-A (Geneart, Regensburg, Germany). All proteins were expressed with C-terminal V5 and His-tags, and were purified by nickel affinity chromatography. Sequences are given in Supplementary Table 3.

**Crystallization of a complex of CD36 and MCvar1 CIDRα2.8.** The purified CD36:CIDRα2.8 complex was concentrated to 10 mg/ml in the presence of 1% (v/v) Flavobacterium meningosepticum endoglycosidase-F1 and carboxypeptidase Y (Sigma-Aldrich) for in situ de-glycosylation and proteolysis. The protein samples were then subjected to sitting drop vapour diffusion crystallization trials in SwissSci 96-well plates by mixing 100 nl protein with 100 nl reservoir solution. The CD36:CIDRα2.8 complex crystallized in 0.2 M NaCl, 20% (w/v) PEG6000, 0.1 M Tris, pH 8.0 at 18°. For cryo-protection, crystals were transferred into mother liquor supplemented with 25% (w/v) glycerol and then cryo-cooled in liquid nitrogen for storage and data collection.

**Data collection and structure determination.** Data were collected at beamline IO2 (Diamond Light Source UK) with a Pilatus 6M detector. Diffraction data of the CD36:CIDRα2.8 complex were indexed and integrated using XIA2 (ref. 50) coupled with XDS[51], and scaled and merged using Aimless[52] to a resolution of 2.07 Å. A subset of 5% of randomly selected diffraction data were used for the calculation of $R_{free}$.

The structure of the CD36:CIDRα2.8 complex was determined by molecular replacement in PHASER[53] with the structure of LIMP-2 (PDB 4F7B) used as a search model to obtain the initial phases. The models were completed by manual building in COOT[54] and refinement in autoBuster[55].

**Surface plasmon resonance.** SPR binding measurements were performed using a Biacore T200 instrument (GE Healthcare Life Science) at 25 °C in 20 mM HEPES, pH 7.5, 0.15 M NaCl, 0.005% (w/v) Tween20. CD36 wild-type and mutants with BAP tags were expressed in HEK293T cells, incubated with BirA for in vitro biotinylation and immobilized onto sensor CAPture chip (GE Healthcare Life Sciences) to ~700 RU. MCvar1 CIDRα2.8 domain and mutants used as analytes were expressed in HEK293 cells and purified as described above. In all binding experiments, analytes were injected for 240 s with a 300 s dissociation time before chip regeneration. Affinities were estimated using equilibrium binding measurements and data were fitted with the BIAevalution software using a one site-binding model. For competition experiments, different CIDRα proteins were injected onto the chip with 500 s association time, followed by injecting either 400 nM Recombinant Human Thrombospondin-1 (TSP-1; R&D SYSTEMS) or 10 nM Native Human low density lipoprotein (oxidized)(oxLDL; AbD Serotec). All buffers were supplemented with 72.5 µM Oleic acid (MP Biochedicals Australasia Pty Limited) for oxLDL binding experiments.

**Size-exclusion chromatography multi-angle light scattering.** SEC-MALS experiments were performed using an analytical Superdex S200 10/300 GL column (GE Healthcare Life Sciences) connected to online static light-scattering (DAWN HELEOS 8, Wyatt Technology) and differential refractive index (Optilabr EX, Wyatt Technology). Purified sample (MCvar1 CIDRα2.8 at 1.5 mg/ml and CD36:MCvar1 CIDRα2.8 complex at 1 mg/ml) were injected into a column

equilibrated with 10 mM HEPES pH 7.5, 0.15 M NaCl. Molecular mass determination was performed using an adapted RI increment value ($dn/dc$ standard value; 0.186 ml/g) to account for glycosylation. The theoretical molecular weight was predicated from amino acid sequence plus 1865 Da per $N$-linked glycosylation site for recombinant protein produced from HEK293T cells in the presence of kifunensine with limited glycosylation. Data were analysed using the ASTRA 6.1 software package (Wyatt Technology).

**Analysis of PfEMP1 sequences.** A total of 263 CIDRα2-6 domains from seven previously annotated whole genome sequenced parasites[15] were used as previously described[44], to blastp extract CIDRα sequences from assemblies of Illumina whole genome sequencing data from 226 parasite samples collected in both Africa and Asia[56] (Study number ERP000190) available through the MalariaGEN community. Only full length domain sequences were kept (domain boundaries as defined in[15]). This resulted in a total of 2386 CIDRα2-6 sequences, including 11 sequences from *P. reichenowi* CIDRα2-6 domains, which were used for the analysis of sequence diversity. A hand-corrected MUSCLE alignment was used to generate sequence logos by WebLogo 3 (ref. 57) and neighbour joining sequence distance trees by MEGA[58]. No major sequence clusters could be verified by bootstrapping, but sequences distributed as previously observed[15].

**Determination of unknown ligand(s) using GC/MS.** To identify the nature of the bound ligand in crystals of the CD36:CIDRα complex, we used gas chromatography coupled with mass spectrometry (GC/MS). Briefly, purified CD36:CIDRα protein was incubated at 100 °C for 5 min, and then chloroform was added to a final concentration of 80% (v/v) to denature the protein and extract the ligand. The sample was derivatized with N,O-Bis(trimethylsilyl)trifluoroacetamide. The derivatized sample was then analysed by GC/MS on a Waters GCT Classic instrument. Ligand identification was performed by comparison of mass spectrum of the analyte with the NIST library database.

**Data availability.** The structure is deposited with pdb code 5LGD. All other data, including CIDRα domain sequences, are available from the authors on request.

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

## Acknowledgements

This work was supported by a Wellcome Trust Investigator Award to MKH. F.-L.H. is supported by a fellowship from the Taiwan Bio-Development Foundation. L.T. and T.L. received support from Danish Council for Independent Research, Sapere Aude program DFF-4004-00624B and The Lundbeck Foundation. We thank David Staunton for help with biophysical methods, the beamline scientists at Diamond Light Source for assistance with data collection, Jakob Jespersen for assistance with sequence analysis, Frank Lennartz for valuable discussion and comments on the manuscript and the mass spectrometry research facility at the University of Oxford. We are grateful for use of data from the MalariaGEN community project on *Plasmodium falciparum* population genomics as described[56].

## Author contributions

F.-L.H. conducted protein production, crystallization and biophysical experiments. F.-L.H. and M.K.H. determined crystal structures. J.R.B. and C.V.R. conducted and analysed mass spectrometry experiments. L.T. produced variant proteins and conducted binding studies. T.L. conducted sequence analysis. F.-L.H., T.L. and M.K.H. wrote the manuscript.

## Additional information

**Competing financial interests:** The authors declare no competing financial interests.

