## [Peer Review File · Nature Communications]

NCOMMS-16-10584

Reviewers' comments:

Reviewer #1 (Remarks to the Author):

In this manuscript by Hsieh et al the authors describe a structure at 2.07Å resolution of the ectodomain of CD36 (residues 35-439) and the MCvar1 CIDR α 2.8 domain (residues 576-158) of the major virulence protein of *P. falciparum*.

They show that CD36 adopts an open conformation, while the MCvar1 CIDR α 2.8 domain adopts a compact structure that is very different conformation from a previous structure obtained for the domain alone. They show that CD36 exhibits a hydrophobic cavity that spans most of the length of the molecule that contains palmitic or stearic acid, with the carboxylic acid stabilised by hydrogen bonding interactions with T195. A likely entry point for the membrane distal fatty acid was identified as well as a second opening is also found at the membrane distal side of the CD36 ectodomain.

A key interaction between CD36 and CIDR α 2.8 is mediated by the insertion of a phenylalanine residue (F153) from CD36 into a hydrophobic pocket on the CIDR α 2.8 domain. The PfEMP1 CIDR α 2.8 domain binds to the α -helical bundle at the membrane-distal tip of CD36, close to the major entrance to the hydrophobic cavity.

Relevant CIDR α s use a triple helix bundle (with an insertion between the second and third core helices) to bind to both CD36 and EPCR, but the binding mechanism shows a structural inversion. The CIDR α 1 domains have a phenylalanine residue (F656) on a convex surface of the domain that protrudes into the hydrophobic groove of EPCR. In contrast, the CIDR α 2 domains have a hydrophobic pocket that binds to a protruding phenylalanine residue (F153) from CD36.

The CD36 interaction site reduces the exposure of chemically conserved determinants from detection by acquired immunity, allowing a diverse set of CIDR α 2-6 domains and a system of antigenic variation to provide the changes necessary to maintain the capacity to cytoadhere and to protect the parasite from splenic clearance.

The authors show that the critical residue for CIDR α 2 binding (F153) is also important for binding to oxidised LDL particles, and that the CIDR α domains prevent oxidised LDL from binding.

This is a very important study that will contribute significantly to our understanding of the virulence processes in *P. falciparum*. I have no major criticisms.

Reviewer #2 (Remarks to the Author):

Hsieh et al report the structure of CD36 in complex with fatty acids and with the CIDR binding region from the malaria parasite. CD36 binds diverse ligands and the new co-crystal structure provides important new molecular insight into both physiological and pathologic binding properties of CD36. The structure shows that CIDR domains engage the oxLDL binding site on CD36, which may limit host options for mutation. Using a combination of site directed mutagenesis and a very comprehensive bioinformatic analysis of CIDR sequences in genome databases, this study clarifies how malaria parasites have retained a CD36 binding capacity, despite extensive diversification of CIDR domains. They show the CIDR binding pocket is conserved in amino acid character, but has highly diversified in sequence. Overall, this study was very carefully done and illuminates an interesting example of parasite-host co-evolution. However, there are few pertinent observations on the malaria parasite-CD36 interaction in the literature that would be worthwhile to discuss in light of the new structure.

1) Other literature observations CD36-parasite interaction that should be considered.

a) Ectophosphorylation of CD36 at threonine 92 regulates both the cytoadherence of *Plasmodium falciparum* IRBCs (Yipp BG et al. Blood 2003; Ho M et al. Inf Immun 2005) and the native CD36 ligand thrombospondin (Asch AS et al. Science 1993). Where is the T92 residue located in the CD36-CIDR co-crystal, and does the new structure give any molecular insights into how dephosphorylation of this residue increases binding?

b) A three amino acid modification (DIE to GHR) or (EIK to GHR) in the M2 region of CIDR was implicated in CD36 binding (Gamain et al Blood 2001). Where are the EIK residues located in the co-crystal and does it give any molecular insights into the binding reduction?

c) *P. falciparum* cytoadhesion to CD36 is strongly enhanced at lower pH (between 6.8 and 7.2). Does the new structure give any molecular insight into the pH-dependent binding enhancement?

2) Lines 186-190 - CD36 adopts an "open conformation", poised to accept fatty acids at neutral pH. It would be helpful to clarify how the "open conformation" in Fig. 1B relates to putative entrance 1 and entrance 2 in Fig. 1C. Is the structure "open" near entrance 1? What is the extent of the "open conformation"?

3) Lines 370-382 - It was somewhat difficult to follow how the CD36 hydrophobic binding cavity is formed in CIDR domains. From supplemental Fig. 7, CD36 contact residues are present in core helices 1

and 2, as well as a polymorphic loop and a three-helical insertion (numbered alpha helices 3- 5) between core helices 2 and 3. Two critical contact residues, F645 and D650, are present in helix 2 and an adjacent loop. Thus, CD36 contact residues are dispersed across the first two-thirds of the domain, but most are not conserved (Fig. 4 and S7).

To help orient the reader, it would be helpful to use light pink and dark pink in Fig. 2 to distinguish between the core helical bundle and the insertion between helices 2 and 3, similar to Fig. 3. This will make it easier to connect Figs. 2, 3, and S7, and understand how the CD36 binding interface is created. It was also a little unclear whether the polymorphic loop that precedes the alpha helical 3-5 insert should be considered part of the hydrophobic binding cavity, since it also contributes contact residues.

Minor comments

- 4) Line 122 - Should mention that CD36-PfEMP1 interaction has been studied with CD36 peptides (Baruch et al. Blood 1999)
- 5) Line 130 - should qualify that CD36 binding "may" benefit the parasite by modulating dendritic cell function. Elliot et al Inf Immun 2007 reported that dendritic cell inhibition was dose dependent and did not depend on PfEMP1.
- 6) Line 199 - should say Fig. S4.
- 7) Line 274 - The CD36 binding interface appears to be more complex than the EPCR binding interface, which is mostly contributed by the ~60 residue insertion. Thus, it is somewhat an oversimplification to say the homology blocks act as the ligand-binding platforms.
- 8) Line 244 - you may want to use the alpha helical numbering from Fig. S7 when referring to the insertion. This will make it clearer when describing the CD36 binding surfaces and hydrophobic cavity.
- 9) Line 412 - The binding surface is surrounded by flexible and sequence diverse loops. However, based on Figs. 2 and S7, there appears to be one loop in the vicinity of the binding surface. Please clarify.
- 10) Lines 441-43 - It is dubious that CIDR domains would make a good drug for preventing the uptake of fatty acids/oxLDL by CD36. Atherosclerosis is a chronic condition and the high antigenicity of CIDR domains would preclude them from being very useful due to rapid clearance by antibodies. If the CD36-lipid interaction is truly an important drug target than why not target it with conventional small molecule drugs?
- 11) Line 572 - It would be good to add that the "CIDR sequences are available from the authors upon request".
- 12) Should be D650 (not F650) in Fig. 2 legend and D680 (not F680) in Fig. S5 legend.

- 13) Fig 4A: Is there bootstrap support for a3.4 and a3.5? What does a2.3/5-11 mean and is there bootstrap support for all of these sequence types?
- 14) Fig. 4A: A) This is a nice figure, it would be good to direct the reader to Fig. S7 in the figure legend, since it is larger font and shows the secondary structure. B) substitute "underlined" for "underplayed" and add that asterisks indicate contact CD36 contact residues in Fig. 4 legend.
- 15) Table S1 - There are some strange symbols under "Data collection"

Reviewer #3 (Remarks to the Author):

This manuscript describes an X-ray crystallographic study of the interaction between CD36 and the CIDR1 domain of the major Plasmodium falciparum cytoadherence factor, PfEMP1. While different regions of PfEMP1 proteins, of which there are >60 in most P. falciparum strains, can bind to different host cell receptors, CD36 binding is conserved amongst most PfEMP1 proteins. The solution of the crystal structure of one such interaction, presented here, is therefore an important milestone for the field. Importantly, the authors go beyond a single structure, and use phylogenetic approaches to establish whether CD36 binding is conserved across the full breadth of P. falciparum diversity, and attempt to define the structural basis for that conservation. The study is meticulous, and the manuscript is well written. Only a few minor modifications are suggested to improve this significant work further.

Major questions.

1. Domain selection.
 - a. Figure 4 shows phylogenetic analysis of CIDR sequences from >200 genomes, but how were those genomes selected? Do they represent the full geographic diversity of P. falciparum? Such spread would be preferable if the authors are attempting to say something truly definitive about functional conservation.
 - b. How were the individual members of each CIDR sub-clade selected for expression? Randomly, or on some other basis? Again, explaining the rationale would help underline whether the findings are truly generalizable.

2. Sequence logo definition. The question of how binding specificity is maintained, despite the extensive diversity between *P. falciparum* genomes, is one of the most important questions in PfEMP1 biology. The experiments investigating this diversity are therefore critical. Figure 4D defines a logo for the residues that make contact with CD36 that appears to have some conserved features. Experimental validation and further investigation of this logo would provide answers to the question of functional conservation for the first time.

a. Presumably, the logo defined in Figure 4D are also the asterisked residues in Figure 4B? This is not currently stated in the legend.

b. It would be helpful to highlight the conserved residues within this logo on a close-up of the CIDR-CD36 interface, such as in Fig 5B.

c. It is hypothesised that it is the limited chemical conservation of the logo shown in Fig 4D that allows conservation of binding. This needs to be functionally tested, on multiple of the different domains expressed in Fig 4A, by mutating the conserved residues and testing the impact on binding by SPR. If the almost completely conserved aromatic residues (582, 586 and 645 in in the MCvar1 sequence) are also universally required for CD36 interaction, then a clear functional motif has been defined and the title of the manuscript proven. This would put the final touch on an otherwise very comprehensive manuscript.

Minor questions

1. Extent of conservation. Recent work on Plasmodium parasites from apes make it clear that EMP1 proteins are not specific to *P. falciparum*, with closely related molecules found much deeper in the ape Plasmodium phylogeny. Have the authors included sequences from these related species in alignments such as those presented in Figure 4A, and tested any of the ape Plasmodium sequences for binding of to CD36? Given the deep conservation of CD36 binding across *P. falciparum*, it would be very interesting to test whether CD36 binding is also conserved beyond the human-infective species. This not essential to the central thrust of the manuscript, but if accomplished it would be obviously complementary, given that the authors are attempting to define conserved features of CD36 binding.

Reviewer 1:

We thank this reviewer for an extremely positive review of the manuscript. No issues were raised in this review that called for a response.

Reviewer 2:

We thank the reviewer for a positive response to our manuscript and also for a careful and helpful analysis. The suggestions, clearly backed by strong insight into the literature, were very useful in producing an improved revision of the manuscript.

- 1) The reviewer draws our attention to the literature on the phosphorylation of T92 of CD36 and to the reduced binding of infected erythrocytes to phosphorylated CD36. The electron density for T92 is clear in our structure and there is also no density due to phosphorylation. However, the side chain of T92 is $>40\text{\AA}$ from the CIDR domain, meaning that any reduction in binding of pRBCs to phosphorylated CD36 is not due to direct inhibition by a blocking phosphate. We are not able to comment further on this issue, but have cited Ho et al (2005) and added two sentences to the end of the first paragraph of page 6 to discuss this point.

The reviewer asks about the findings of Baruch et al, who identified a three amino acid sequence which allowed them to distinguish between CD36 binding and non-binding CIDR domains. These three residues do not make

direct contact with CD36 but are closely placed to the ligand in the homology block that forms the majority of the binding surface. The presence of these residues in the homology block that interacts with CD36, rather than a role in directly contacting CD36, might explain their association with CD36-binding PfEMP1. We have mentioned this in a new sentence in the first paragraph of page 7 and have added this reference.

The reviewer then raised the issue of the pH dependence of binding to CD36, as it has been demonstrated that binding of infected erythrocytes to CD36-expressing COS cells is strongest at a pH of ~6.8. Sadly our structure does not give much insight into this. A small number of interactions are mediated by hydrogen bonds (see Table S2), which might be modulated by pH changes, but this is unlikely to have a major effect. It is also possible that subtle conformational changes take place on a change in pH, although, as already discussed, we think it unlikely that there will be major conformational shifts as seen in LIMP-2 due to the absence of the pH-sensor histidine. As pH-dependence has been shown in an assay in which a cell binds to a cell, there are many possible reasons for these pH effects, including changes in the surface display of the receptors or their arrangement and the change in the binding level might not be due to a change in the affinity of the protein-protein interaction.

- 2) In terms of the entrances to the CD36, it is entrance 1 which corresponds to the entrance in LIMP-2 that opens and closes in response to pH changes. We have clarified this through the addition of a clause in the second paragraph of page 4. We have also altered the first paragraph of page 7 to clarify. We have changed the first paragraph of page 8 to indicate which of these interacting residues are most conserved. We have also changed the second paragraph of page 9 to clarify the contribution of the core bundle and the inserted region in forming the hydrophobic pocket.
- 3) We agree with the reviewer that the description of the hydrophobic cavity was not as clear as it should have been! We have edited Figure 2B as suggested, to colour core helices light pink and the insert dark pink. We have also labelled more of the interacting residues on this diagram so that a reader can see which residues form the hydrophobic pocket and can compare their location with the sequence in Supplementary Figure 7.
- 4) We agree with the reviewer that it is right to mention the peptide study of Baruch that identified residues 146-164 of CD36 as containing the PfEMP1 binding site. Indeed our findings agree with this study. In addition to mentioning this work at the location suggested, we have credited this discovery in the last paragraph of page 5.
- 5) We have added 'may' as suggested in response to the comments on dendritic cell modulation, and have cited Elliot et al (2007).
- 6) We have made this correction.

- 7) We agree with the point about our use of the phrase 'ligand-binding platforms' and have changed this to 'contribute to ligand-binding sites'.
- 8) We have clarified our description of the regions of the CIDR α 2.8 domain that contact CD36 by addition of the residue numbers for the α -helix and loop described here. We have also used the helix numbers and referred to Supplementary Figure 7 when describing the 'core' helices a few paragraphs later.
- 9) The reviewer is right to point out that this sentence was not clear and that only one loop makes direct contact to CD36. We have rewritten this sentence to clarify.
- 10) We agree with the reviewer that CIDR domains are highly unlikely to be effective atherosclerosis medicines. We have added a clause to clarify that we intended our discussion to suggest using them to study the effect of targeting this region of CD36 rather than as a putative candidate for therapeutic use.
- 11) We have added the sentence as suggested.
- 12) We thank the reviewer for their careful reading of these legends and have corrected these errors.
- 13) The question about bootstrap support for the different groups of CIDR α domains is a good one. The sequence diversity and number of sequences renders each branch poorly defined and there is no bootstrap support for most of these groups (apart from CIDR α 2.4, CIDR α 6, and smaller subsets). Despite this, the tree reflects the best 2D representation of the relative sequence similarity between the domains and the same distribution is seen in trees derived from different datasets. In support of this, trees derived from the CIDR α domains from seven genomes (Rask et al, 2010) show the same clustering distribution. To clarify this, we have added 'No major sequence clusters could be verified by bootstrapping' to paragraph three of page 7 and we have clarified this point in the methods section. We have also altered the annotation of the figure to clarify the meaning of α 2.3/5-11.
- 14) We have added a link to Supplementary Figure 7 and made the other suggested changes.
- 15) This has been corrected – these symbols appeared in the pdb conversion on submission and went unnoticed!

Reviewer 3:

We thank the reviewer for their supportive comments and for recognising our work as an important milestone. We are grateful for the helpful suggestions about how to clarify the breadth of our findings.

The reviewer first asks about our selection of genomes for the bioinformatics analysis. These genome sequences represent parasites from all continents (Rask et al 2010, Manske et al 2012) albeit with an over representation of samples from African and Asia. As described in Rask et al 2010, there is no clustering of sequences from Africa vs Asia vs South America. The sequences can therefore be expected to represent a random sampling of the global diversity of PfEMP1 and are the most representative set available at present. We have added a few words to the third paragraph of page 7 to make this point.

The reviewer asks about the selection of proteins for binding studies. We selected the MCvar1 CIDR α 2.8 domain for structural studies as a previous structure of the domain revealed it to have proven crystallization properties (Klein et al 2008). The proteins analysed for CD36 binding were chosen randomly from around the tree to probe the most sequence diverged groups without bias. As described in the response to reviewer 2, the tree does not show strong clustering of subtypes indicating no functional specialization within the group of sequences. Therefore a random testing across the tree is the right way to achieve an unbiased assessment of CD36 binding capability. We have added a phrase to the third paragraph of page 7 to make this clear.

We have clarified in the figure legend to Figure 4B that the starred residues make direct contacts with CD36.

We agree that it would be helpful to have a figure showing the location of the most conserved residues on the CIDR α domain involved in binding to CD36. These residues were actually shown in Figure 2B but had not been labelled. We have now altered Figure 2B to include labels for these residues and have referred back to this figure at the end of the paragraph in which we discuss conservation (the first of page 8).

The reviewer also raises an important question about the conservation of binding by the different groups of CIDR α 2-6 domains. The chemical conservation of the interacting residues does, as the reviewer points out, suggest that the binding pocket will be conserved. We have now added some additional data to support this conclusion. Rather than making point mutants of each of the different CIDR α domains (which would take at least six months and a significant investment of resource, as these domains are all expressed in eukaryotic expression systems) we have used the F153A mutant of CD36. We show that this mutation has a significant effect on the binding of all of the CIDR α domains to CD36, and abolishes the binding of the majority at the concentrations tested. This data is now presented in Figure 5C and is described in a new third paragraph of page 8. Together with the conservation of residues in the binding pocket, this provides strong support for conservation of binding and we believe provides the supporting evidence that the reviewer requests.

The review also asked about EMP1 proteins from apes. While (as acknowledged) this is a separate question from that answered in this study, eight CIDR sequences from *P. reichenowi* EMP1 proteins had been included in the analysis.

We have clarified this in the methods section in the paragraph 'comparison of sequence'. These sequences behave similarly to those from *P. falciparum*. The PrEMP1 sequences do not form distinct sequence clusters or long divergent branches but are spread within the tree. We would therefore predict them to bind to CD36. It will be interesting for a future study on the evolution of the PfEMP1 to follow this up with binding studies.

In summary, we thank the reviewers again for their extremely positive response to our manuscript and trust that it is now be ready for publication.

With best wishes,

NCOMMS-16-10584A

REVIEWERS' COMMENTS:

Reviewer #2 (Remarks to the Author):

The revised manuscript fully addresses my comments. I have only very minor suggestions.

Minor comments

1. As clarification, pH dependence is also observed with purified CD36 protein from platelets (Craig et al. *Inf Immn* 1997 65:4580-85), and is not simply a cell-based phenomenon. However, it is fine if the new co-crystal structure does not give any mechanistic insight into this phenomenon.
2. Discussion, paragraph 2 - The "insertion" in CIDRa2-6 domains is described as two shorter helices in the discussion. However, it appears to be three alpha helices with a bend between the α_4 and α_5 helices in Fig. 3 and Fig. S7. This bend is the location of the concave hydrophobic pocket. You may want to clarify.
3. Fig. S2 - provide the PDB accession numbers for the CIDRa1 and CIDRg structures.

Reviewer #3 (Remarks to the Author):

The revised version of this already excellent manuscript has been further improved and clarified, and all major questions dealt with.

A minor comment only - the fact that *Plasmodium reichenowi* sequences were included in the analysis is mentioned in the Methods, but the outcome of that analysis (that these ape sequences also contain conserved CD36 binding motifs) is currently mentioned in the rebuttal letter, but not in the manuscript text. This is an interesting finding, worthy of a couple of sentences in either the Figure 4 legend or page 7/8. It is not insignificant - the authors give much importance, quite rightly, to the fact that CD36 binding is maintained despite extensive sequence diversification across *Plasmodium falciparum*. The inclusion of

the *P. reichenowi* sequences, and the identification of a binding motif within them, suggests that CD36 binding is even older than *Plasmodium falciparum*, and has been conserved despite sequence diversification across different *Plasmodium* species. Completely up to the authors whether to make this point more clearly, but given that Nature Communications has published several manuscripts on ape *Plasmodium* in general, and var sequences in particular, it would seem apposite to publish this finding here as well.

REVIEWERS' COMMENTS:

Reviewer #2 (Remarks to the Author):

The revised manuscript fully addresses my comments. I have only very minor suggestions.

Minor comments

1. As clarification, pH dependence is also observed with purified CD36 protein from platelets (Craig et al. *Inf Immn* 1997 65:4580-85), and is not simply a cell-based phenomenon. However, it is fine if the new co-crystal structure does not give any mechanistic insight into this phenomenon.
2. Discussion, paragraph 2 - The "insertion" in CIDRa2-6 domains is described as two shorter helices in the discussion. However, it appears to be three alpha helices with a bend between the a4 and a5 helices in Fig. 3 and Fig. S7. This bend is the location of the concave hydrophobic pocket. You may want to clarify.
3. Fig. S2 - provide the PDB accession numbers for the CIDRa1 and CIDRg structures.

In response to reviewer 2:

- (i) We have clarified what we described as the 'two α -helices' in the discussion by adding a few words to refer back to the helix numbers in Supplementary Figure (now) 8. The referee is right in that there are three α -helices in this inserted region. However only two of them match the equivalent helices of the EPCR-binding CIDR α 1 domains and so only two are discussed here. We hope that the addition of helical numbering clarifies which helices we are discussing.
- (ii) We have added the pdb accession numbers to the legend for Supplementary Figure (now) 3.

Reviewer #3 (Remarks to the Author):

The revised version of this already excellent manuscript has been further improved and clarified, and all major questions dealt with.

A minor comment only - the fact that *Plasmodium reichenowi* sequences were included in the analysis is mentioned in the Methods, but the outcome of that analysis (that these ape sequences also contain conserved CD36 binding motifs) is currently mentioned in the rebuttal letter, but not in the manuscript text. This is an interesting finding, worthy of a couple of sentences in either the Figure 4 legend or page 7/8. It is not insignificant - the authors give much importance, quite rightly, to the fact that CD36 binding is maintained despite extensive sequence diversification across *Plasmodium falciparum*. The inclusion of the *P. reichenowi* sequences, and the identification of a binding motif within them, suggests that CD36 binding is even older than *Plasmodium falciparum*, and has been conserved despite sequence diversification across different *Plasmodium*

species. Completely up to the authors whether to make this point more clearly, but given that Nature Communications has published several manuscripts on a *Plasmodium* in general, and var sequences in particular, it would seem apposite to publish this finding here as well.

In response to reviewer 3:

We have now added a little more information to the main manuscript about the *Plasmodium reichenowi* sequences. We have indicated these sequences as red lines in Figure 4A and have described this in the figure legend. We have also added the sentence 'the alignment included 11 CIDR α domains from *Plasmodium reichenowi* that distributed across the tree of *Plasmodium falciparum* CIDR α 2-6 domains, as seen for other surface protein families⁴⁷' to the main text. This should allow readers interested in the evolution of the *Laverania* species to see how the *Plasmodium reichenowi* sequences are distributed across the tree of sequence diversity.

We believe that we have made all of the suggested changes and hope that the manuscript is now in a final draft and is ready for publication.